# Resistance to Triazoles in Populations of *Mycosphaerella fijiensis* and *M. musicola* from the Sigatoka Disease Complex from Commercial Banana Plantations in Minas Gerais and São Paulo, Brazil

**DOI:** 10.3390/microorganisms13071439

**Published:** 2025-06-20

**Authors:** Abimael Gomes da Silva, Tatiane Carla Silva, Silvino Intra Moreira, Tamiris Yoshie Kiyama Oliveira, Felix Sebastião Christiano, Daniel Macedo de Souza, Gabriela Valério Leardine, Lucas Matheus de Deus Paes Gonçalves, Maria Cândida de Godoy Gasparoto, Bart A. Fraaije, Gustavo Henrique Goldman, Paulo Cezar Ceresini

**Affiliations:** 1Department of Crop Protection, Agricultural Engineering and Soil, São Paulo State University, UNESP, Ilha Solteira 15385-000, SP, Brazil; ag.silva@unesp.br (A.G.d.S.); tamiris.kiyama@unesp.br (T.Y.K.O.); macedo.souza@unesp.br (D.M.d.S.); gabrielaleardinevl@gmail.com (G.V.L.); lmdp.goncalves@unesp.br (L.M.d.D.P.G.); 2Faculty of Pharmaceutical Sciences of Ribeirão Preto (FCFRP), University of São Paulo, USP, Ribeirão Preto 14040-900, SP, Brazil; tatiane.carla@unesp.br (T.C.S.); ggoldman@usp.br (G.H.G.); 3Institute of Agricultural Sciences, Federal University of Uberlândia, Uberlândia 38400-902, MG, Brazil; silvinointra1@gmail.com; 4Faculty of Agricultural Sciences from Ribeira Valley, São Paulo State University, UNESP, Registro 11900-000, SP, Brazil; maria.candida@unesp.br; 5Business Unit Biointeractions & Plant Health, Wageningen University & Research, 6700 AA Wageningen, The Netherlands; bart.fraaije@wur.nl; 6National Institute of Science and Technology in Human Pathogenic Fungi, Ribeirão Preto 14040-900, SP, Brazil

**Keywords:** propiconazole, tebuconazole, demethylase inhibitor fungicides (DMIs), chemical control

## Abstract

The sterol demethylation inhibitors (DMIs) are among the most widely used fungicides for controlling black Sigatoka (*Mycosphaerella fijiensis*) and yellow Sigatoka (*Mycosphaerella musicola*) in banana plantations in Brazil. Black Sigatoka is considered more important due to causing yield losses of up to 100% in commercial banana crops under predisposing conditions. In contrast, yellow Sigatoka is important due to its widespread occurrence in the country. This study aimed to determine the current sensitivity levels of *Mf* and *Mm* populations to DMI fungicides belonging to the chemical group of triazoles. Populations of both species were sampled from commercial banana plantations in Registro, Vale do Ribeira, São Paulo (SP), Ilha Solteira, Northwestern SP, and Janaúba, Northern Minas Gerais, and were further characterized phenotypically. Additionally, allelic variation in the *CYP51* gene was analyzed in populations of these pathogens to identify and characterize major mutations and/or mechanisms potentially associated with resistance. Sensitivity to the triazoles propiconazole and tebuconazole was determined by calculating the 50% inhibitory concentration of mycelial growth (EC_50_) based on dose–response curves ranging from 0 to 5 µg mL^−1^. Variation in sensitivity to fungicides was evident with all nine *Mf* isolates showing moderate resistance levels to both propiconazole or tebuconazole, while 11 out of 42 *Mm* strains tested showed low to moderate levels of resistance to these triazoles. Mutations leading to CYP51 substitutions Y136F, Y461N/H, and Y463D in *Mm* and Y461D, G462D, and Y463D in *Mf* were associated with low or moderate levels of resistance to the triazoles. Interestingly, Y461H have not been reported before in *Mm* or *Mf* populations, and this alteration was found in combination with V106D and A446S. More complex CYP51 variants and *CYP51* promoter inserts associated with upregulation of the target protein were not detected and can explain the absence of highly DMI-resistant strains in Brazil. Disease management programs that minimize reliance on fungicide sprays containing triazoles will be needed to slow down the further evolution and spread of novel CYP51 variants in *Mf* and *Mm* populations in Brazil.

## 1. Introduction

The banana Sigatoka disease complex (BSDC) encompasses black Sigatoka (caused by *Mycosphaerella fijiensis*, *Mf*) and yellow Sigatoka (caused by *M. musicola*, *Mm*). Black Sigatoka was initially reported in Brazil in 1998 in the Amazon region and has since been detected in 19 states, including São Paulo and Bahia, the country’s primary banana-producing states [1,2,3,4,5]. This disease is a significant constraint to banana production, capable of reducing yields by up to 100%. However, there are reports suggesting that black Sigatoka may have been misidentified as the less devastating yellow Sigatoka, which was first identified in the Amazon region in 1944 and is widespread across all banana-growing regions of Brazil, causing yield losses of up to 50% [6,7,8,9].

The BSDC diseases are polycyclic, with pathogens *Mf* and *Mm* exhibiting a mixed reproductive system enabling dispersal via asexual conidia over short distances by rain splash and sexually produced ascospores over long distances by air [10,11]. Populations of *Mf* and *Mm* in Brazil, Mexico, and the Philippines show high genotypic variation due to sexual reproduction and gene flow from distant pathogen migration [6,8,12,13]. Genetic resistance to BSDC is generally absent or partial in most commercial banana cultivars, necessitating disease control strategies primarily based on calendar-based systemic or protectant fungicide sprays [6].

In regions with high disease pressure, extensive fungicide applications, up to 52 sprays of protectant or 26 sprays of systemic fungicides per year, are applied, leading to increased production costs and environmental impact [5,6,14]. Conversely, in the Ribeira Valley of Brazil, black Sigatoka control involves weekly disease monitoring and frequent fungicide applications, resulting in 15–20 sprays per year [5]. Control heavily relies on systemic site-specific quinone outside inhibitor (QoI), demethylation inhibitor (DMI), and the more recently introduced succinate dehydrogenase inhibitors (SDHIs) fungicides. However, this intensive use of fungicides exerts selective pressure on pathogen populations, leading to the emergence, selection, and spread of fungicide-resistant strains [15,16,17,18,19,20].

Several studies on fungicide resistance in populations of *Mf* and *Mm* from Brazil have been carried out. Concerning QoIs, the first surveys in Brazil initially showed no resistance in *Mf* populations from the Amazon and Ribeira Valley in São Paulo, as well as in *Mm* populations from the Federal District and São Paulo sampled from 2008 to 2018 [6,21,22]. However, recent data indicate varying levels of QoI resistance, with 10.0% and 9.4% of *Mf* and *Mm* isolates, respectively, carrying the G143A substitution in cytochrome *b*, particularly in Southeastern Brazil (São Paulo and Minas Gerais), a region with an intensive fungicide use [14,19].

Similarly, resistance to SDHI fungicides has also been detected in *Mf* and *Mm* populations in Southeastern Brazil sampled during 2020–2021, a few years after its labeling [20]. Globally, SDHI fungicides approved for BSDC management in banana plantations include boscalid, fluopyram, fluxapyroxad, and isopyrazam [23]. In Brazil, only one SDHI fungicide co-formulation (Collis™ from BASF) with the SDHI boscalid and the QoI kresoxim-methyl is labeled for BSDC [20]. Some fungal isolates with resistance to boscalid and fluxapyroxad showed Sdh target site alterations (SdhC N55D, SdhB E196Q, and SdhD K66N) [20]. None of these substitutions have been associated with SDHI fungicide resistance in BSDC pathogens, based on the latest survey conducted by the SDHI Working Group in 2022 [23]. Continued monitoring for other mutations and resistance mechanisms such as multiple Sdh paralogs and overexpression of efflux pumps is critical for effective disease control strategies [20].

Regarding resistance to DMI fungicides, there is evidence of reduced sensitivity in *Mf* populations since 2008–2009 [21,22]. The DMI fungicides, introduced in the 1980s, are among the most widely used for controlling Sigatoka diseases [24,25]. DMIs inhibit the biosynthesis of ergosterol, an essential component of the fungal cell membrane [26]. Their target is 14α-demethylase encoded by the *CYP51* gene, a member of the cytochrome P450 family [27]. Studies on plant pathogens have shown that target site mutations appear to be the least predictable for CYP51 [28]. Given the frequent spray of DMI fungicides in banana plantations, the resulting selection pressure may lead to increased frequency of new *Mf* and *Mm* genotypes with reduced sensitivity to these fungicides. Continuous monitoring and detailed investigation into resistance evolution and spread in favorable agroecosystems are crucial for effective disease management and sustainable control strategies [3,10,29,30,31].

Mechanisms of resistance to DMI involving *CYP51* mutations have been reported in *Mf* and *Mm* both elsewhere and in Brazil [15,24], with CYP51 overexpression being linked to tandem repeats in *Mf* [17]. In Brazil, *CYP51* mutations resulting in G462D and Y463H in *Mf* [14] and A381G, Y461N, and Y463H in *Mm* [1,14,24] have been associated with DMI resistance in insensitive strains of the pathogens from the Federal District (Central-western Brazil) and Ribeira Valley region in São Paulo (Southeastern Brazil). A comprehensive study on 266 *Mf* strains originating from the Americas, Africa, and Asia showed that CYP51 substitutions Y136F, A313G, H380N, D460E/V, ΔY461, Y461D/N/S, G462A/D, and Y463D/H/N/S were only present in strains with reduced sensitivity to DMIs [15,17,25]. CYP51 substitutions T18I, D106V, and A446S were frequently found in both DMI-sensitive and -resistant strains. Accumulation of mutations resulting in combinations of CYP51 substitutions were identified. CYP51 variants [V106D, A313G, and D460V], [T18I, V106D, Y136F, and Y463D] and [T18I, V106D, A313G, and Y463D/H/N] in combination with promoter inserts containing tandem copies or palindromic motifs leading to overexpression of the fungicide target protein were most frequently found in insensitive strains [25].

Our study aims to provide updated insights into DMI resistance prevalence and mechanisms in *Mycosphaerella* species associated with BSDC in Brazil, contributing to more sustainable approaches for disease control.

Therefore, based on these premises, the main objectives of our study were: (i) to determine the current sensitivity levels of populations of *Mf* and *Mm* from banana plantations in the Ribeira Valley, in Northwestern São Paulo, and in Northern Minas Gerais (Southeastern Brazil) to DMI fungicides; and (ii) to determine the variation in the *CYP51* gene to identify key mutations and to characterize the mechanisms associated with resistance to DMI fungicides locally.

## 2. Materials and Methods

### 2.1. Sampling of Diseased Plants and Isolation of the Fungal Pathogens

The Brazilian Ministry of Environment/National System for the Management of Genetic Heritage and Associated Traditional Knowledge; SisGen issued Certificates #A64D0EA and A100786 authorizing the scientific activities associated with the collection of botanical and fungal material from the banana agroecosystems in the Cerrado’s and Atlantic Forest’s biomes and access to the genetic diversity of Mycosphaerella species.

During 2020, diseased plants were obtained from commercial banana plantations from Northwest São Paulo (populations designated SPNW-C and SPNW-O), from Ribeira Valley in Southern São Paulo (SPVR-CI), and from Northern Minas Gerais (MGN-C), Brazil. These geographic populations were associated with distinct disease management systems: (i) intensive management (population SPVR-CI, sampled from Jacupiranga, Registro, and Sete Barras counties, in the Ribeira Valley); (ii) reduced management (population SPNW-C, obtained from Ilha Solteira county, in Northwest São Paulo, and MGN-C, sampled from Janaúba, Northern Minas Gerais); and (iii) organic management, with no fungicide applications (population SPNW-O, also obtained from Ilha Solteira).

In Ribeira Valley (SPVR-CI), the management strategy involved intensive fungicide spraying, with 8 to 14 preventive applications annually for chemical control of black Sigatoka [14,19]. This region is renowned as a major center for banana production in São Paulo state and Brazil. The predominant banana cultivars there are the BSDC-susceptible Prata (*Musa* spp. AAB, commonly known as “Lady Finger” banana) and Nanica (*Musa* spp. AAA, Cavendish subgroup) [9,19,31,32]. In the second geographic population, located in Northwestern São Paulo (SPNW-C), and the third site, in Northern Minas Gerais (MGN-C), fungicide use is reduced to four to five preventive sprays targeting yellow Sigatoka [19]. The SPNW-C area encompasses 40 hectares dedicated to the susceptible Maçã variety (triploid AAB) [19]. Conversely, in MGN-C, the banana plantation comprises Prata and Nanica varieties [19]. At the final sampled site (SPNW-O), no fungicides were used. This area includes several small family plantations with banana plants of various varieties and ages in Ilha Solteira county [19].

Fragments (approximately 20 × 20 cm) of banana leaves showing symptoms of either yellow Sigatoka or black Sigatoka diseases from different ages and cultivars were collected. At each location, sampling points consisted of 50 m^2^ areas, where approximately five plants were sampled. The leaf fragments were placed in paper bags and transported to the Molecular Plant Pathology Lab at UNESP, Ilha Solteira Campus. Samples were refrigerated until pathogen isolation.

Pathogen isolation followed the methodology of [24] with modifications. Leaves with apparent disease symptoms were rinsed with running water. Samples taken from the leaves, including the lesion area and its marginal parts; disinfected with a 75% ethanol solution; and then rinsed with distilled water to remove excess alcohol, approximately for a minute each. Subsequently, they were dried on sterilized filter paper and incubated in Petri dishes containing water agar medium (15 g L^−1^ agar supplemented with 50 μg mL^−1^ chloramphenicol and streptomycin). Fungal mycelial growth was analyzed under a stereoscopic microscope to distinguish characteristic sporodochia of *Mf* and *Mm* [19,20]. Direct isolation was performed by transferring conidia from the sporodochia formed in water agar medium to plates containing potato dextrose agar (PDA: 20.8 g L^−1^ potato dextrose and 15 g L^−1^ agar) using the streaking technique, followed by incubation for five days at 25 °C under a 12 h photoperiod [7]. Colonies were purified by transferring pure microcolonies of *Mf* or *Mm* isolates to new PDA plates and incubating for five days under the same conditions.

For long-term preservation of isolates, sterilized 0.5 cm^2^ filter paper fragments were subsequently transferred onto the surfaces of growing fungal colonies, maintained under incubation at 25 °C under a 12 h photoperiod until complete colonization of the fragments. Filter paper fragments colonized by fungal mycelial growth were transferred to Petri dishes and dried under sterile conditions in a laminar flow hood for 24 h. Subsequently, these colonized and dried paper fragments were transferred to cryotubes with silica gel and cryopreserved in a −20 °C freezer. These isolates constitute a subset of the population examined for QoI resistance by [19] and for SDHI resistance by [20].

### 2.2. Molecular Identification of Species

For the molecular identification of the two target species, the isolates were reactivated on PDA medium and grown for 10 days at 25 °C under a 12 h photoperiod. Approximately 1 cm^2^ of mycelial culture per isolate was then lyophilized for 48 h. The lyophilized mycelium was used for DNA extraction using the Wizard^®^ Genomic DNA Purification Kit (Promega, Madison, WI, USA), following the manufacturer’s instructions. The quantification of extracted DNA was performed using a NanoDrop^®^ 2000c spectrophotometer (Thermo Fisher Scientific, USA) to achieve final dilutions at a concentration of 100 ng μL^−1^.

Specific PCR primers were used for the molecular identification of *Mf* (CYP51_Pfijien_F1 and R1) and *Mm* (CYP51A_Mm_F523 and R2457) based on the amplification of the *CYP51* gene (Table 1). Polymerase chain reaction (PCR) reactions were performed using a ProFlex thermal cycler (Applied Biosystems, Carlsbad, CA, USA). Reactions were prepared in a total volume of 25 μL, consisting of 1 × PCR buffer (Applied Biosystems, Foster City, CA, USA), 1.5 mM MgCl_2_, 60 μM dNTPs, 0.2 μM of each primer, 1.5 U Taq DNA polymerase (Invitrogen-Thermo Fisher Scientific, Waltham, MA, USA), and 100 ng of genomic DNA. The PCR conditions for Mf were as follows: initial denaturation at 95 °C for 5 min, followed by 36 amplification cycles with temperatures of 94 °C for 30 s, 58 °C for 60 s, and 72 °C for 60 s, with a final extension at 72 °C for 7 min. For Mm, the conditions were similar, with the only difference being in the annealing temperature, which was 59 °C. The resulting amplicons were analyzed by agarose gel electrophoresis, and the species were identified based on the size of the fragments obtained. Positive identification of *Mf* was confirmed by amplification of a 1700 bp fragment, while for *Mm*, a 1900 bp fragment was amplified. Positive controls for both species and negative controls using DNA extracted from the basidiomycete fungus *Rhizoctonia solani* AG-1 IA and the ascomycete fungus *Pyricularia oryzae Triticum* lineage, available in our laboratory, were included.

### 2.3. Identification of Allelic Variation of the CYP51 Gene

To assess allelic variation of the *CYP51* gene in the *Mf* and *Mm* isolates, specific primers targeting the *CYP51* gene were used for PCR amplification and sequencing. We conducted PCR amplification and sequencing of a total of nine isolates of *Mf* and forty-two isolates of *Mm* from the four BSDC populations sampled. The primers used in this study (Table 1) were either the ones used in other studies [33] or were designed specifically for this study. These particular primers were designed using Primer3 software within Geneious Prime version R 9.0.5 (Biomatters, Auckland, New Zealand). These primers also cover the promoter region of the CYP51 gene, allowing the presence of repetitive elements within the region to be verified, indicated as another mechanism associated with resistance to DMI through upregulation of gene expression [25]. Reference *CYP51* sequences used for primer design were obtained from NCBI/GenBank^®^ or from a specific publication [33], which included: genomic scaffold sequence NW_006921538 and EF581093.1 for *Mf* and MF521833 and MF521834 for *Mm*. These sequences were derived from *Mf* strains CIRAD86 (Cameroon, 2006), Bo_1 (Philippines, 2013), and CaM10-6 (Costa Rica, 2014) and from *Mm* strains 2SJA and 9SJA (Brazil, 2014) [1,10,24,33].

The PCR reactions for amplifying and sequencing the *CYP51* gene were carried out in a final volume of 30 µL, containing water, 50 ng of total DNA, 0.3 µM of each primer, 0.2 mM of each dNTP, 2.5 mM MgCl_2_, 3 µL of 10× buffer, and 0.5 U of Taq DNA polymerase (Sigma-Aldrich, St. Louis, MO, USA). Amplifications were performed using a ProFlex thermal cycler (Applied Biosystems, USA) with an initial denaturation at 95 °C for 5 min, followed by 36 cycles of 94 °C for 30 s, 1 min at the specific annealing temperature for each primer pair (see Table 1), and 72 °C for 1 min, with a final extension at 72 °C for 7 min.

PCR products were purified and Sanger-sequenced at the sequencing facilities of Macrogen Inc. in Seoul, South Korea. To ensure accuracy, four or six complementary sequences were generated for each *CYP51* gene fragment amplified for *Mf* or *Mm*, respectively, which included internal primers (Table 1). DNA sequences were analyzed and aligned using Geneious R 9.0.5 software (Biomatters, New Zealand) to identify alleles, haplotypes, and distinguish non-synonymous mutations leading to amino acid changes in inferred protein sequences. The same reference sequences described previously were used for annotation and derivation of CYP51 protein sequences from the experimental sequence data obtained in our study. The amino acid substitutions in the CYP51 variants detected were depicted as lollipops graphs built using the R software version 4.2.0 libraries *ggplot2*, *dplyr*, *hrbrthemes*, *data.table*, *tidyverse*, *readxl*, *glue*, *ggtext*, and *ggrepel*, and the functions *geom_segment*, *ggtitle*, *geom_point*, and *geom_text_repel* [34].

### 2.4. Phenotypic Assessment of In Vitro Sensitivity to DMI

For the phenotyping of sensitivity to DMI fungicides, a total of 9 isolates of *Mf* and 42 isolates of *Mm* were selected: 9 isolates of *Mf* from SPVR-CI, 15 isolates of *Mm* from MGN-C, 13 isolates of *Mm* from SPNW-C, and 14 isolates of *Mm* from SPNW-O.

Fungicide sensitivity testing was carried out with the triazoles propiconazole and tebuconazole. Formulated propiconazole (Tilt™ EC, active ingredient at 250 g L^−1^, Syngenta S.A., Brazil) was diluted in deionized water to obtain a stock solution of 25 µg mL^−1^. Formulated tebuconazole (Riza™ 200 EC, active ingredient at 200 g L^−1^, FMC Química do Brasil Ltda, Paulínia, Brazil) was diluted to obtain a stock solution of 20 µg mL^−1^. The final tested doses for propiconazole and tebuconazole were 0, 0.001, 0.01, 0.05, 0.1, 1, and 5 µg mL^−1^. The fungicides tested at different doses were mixed with PDA medium supplemented with chloramphenicol and streptomycin, both at a concentration of 50 μg mL^−1^. Sensitivity tests were performed in 90 mm diameter Petri plates.

Mycelial fragment suspensions for the fungicide sensitivity testing were obtained from the fungal colonies grown for 10 days on PDA at 25 °C under dark conditions. Mycelial samples with an area of 1 cm^2^ were collected and macerated in crucibles using ceramic pistils under sterile conditions in a laminar flow hood. The samples were then resuspended in 5 mL of sterilized deionized water containing 0.05 mL^−1^ of Tween-20. Subsequently, 5 µL of the inoculum suspension was carefully pipetted onto 0.5 cm diameter sterile filter paper discs, positioned on top of PDA medium containing different fungicide concentrations. The 90 mm diameter plates were sealed with plastic film to prevent drying and external contamination and incubated for 15 days at 25 °C and a 12 h photoperiod, after which growth was measured.

### 2.5. Experimental Design, Statistical Analyses, and Data Depiction

The experimental design consisted of complete randomized blocks with four replicates per treatment and experiments in duplicate. Sensitivity to the two DMI fungicides was assessed by determining the effective concentration of 50% to inhibit fungal growth (EC_50_, in µg mL^−1^), estimated using a dose–response function implemented in the Excel macro ED50plus v1.0 [35]. For hypothesis testing, isolates were grouped by the CYP51 protein variants detected, by geographical populations of the pathogens, and also by species. Analysis of variance (ANOVA) and the Scott-Knott test (at 5% probability) for means comparison were performed in the R environment using the statistical packages *agricolae* and *ScottKnott* [34]. We also classified the individual isolates according to their sensitivity to the DMI fungicides propiconazole and tebuconazole using the phenotypic classes presented in Table 2. We also determined the correlation between the EC_50_ values for propiconazole and tebuconazole to look for evidence of cross-resistance between the two fungicides.

The boxplot figures depicting the contrast among the CYP51 protein variants detected, among geographical populations of the pathogens, and between isolates grouped by species were built using the R software library tidyverse 1.3.1, which included the packages *ggplot*, purrr, *tibble*, *dplyr*, *tidyr*, *stringr*, *readr*, and *forcats* and the functions *ggplot*, *geom_boxplot*, *stat_summary*, *geom_jitter*, *ggtitle*, *theme*, and *geom_text* (35). The whole set of color palettes chosen to build figures with accessibility features included color-blind-safe and print-friendly colors, using the resources from Color Brewer 2.0 (URL: https://colorbrewer2.org/#type=sequential&scheme=BuGn&n=3, accessed on 1 November 2023).

## 3. Results

### 3.1. Identification of Allelic Variation of the CYP51 Gene

In the present study, we characterized a total of 51 isolates of the BSDC pathogens, comprising 42 from *Mm* and 9 from *Mf*. A total of eight different CYP51 protein variants were detected among these isolates (Figure 1). *Mm* exhibited the highest diversity of protein variants of the *CYP51* gene (N = 5). CYP51 variant A ([V106D and A446S] from *Mm* was most frequently found in the populations tested (N = 31), while variants B [V106D, Y136F, and A446S] (N = 2), C [V106D, A446S, and Y461H] (N = 3), D [V106D, A446S, Y461N, and Y463D] (N = 1) and E [V106D, A446S, and Y463D] (N = 5) were detected at much lower numbers. As for the *Mf* population (N = 9) three CYP51 variants were detected: variant F [T18I, V106D, and Y463D] (N = 5), G [T18I, V106D, and Y461D] (N = 3), and H [T18I, V106D, and G462D] (N = 1) (Figure 1).

CYP51 substitutions V106D and A446S were detected in all *Mm* isolates (N = 42), while substitution T18I was absent in *Mm* but detected in all nine *Mf* isolates. CYP51 substitutions associated with azole resistance were found less frequently. CYP51 Y136F (N = 2), Y461H (N = 3), and Y461N (N = 1) were found in *Mm*, whereas Y461D (N = 3) and G462D (N = 1) were detected in *Mf*. CYP51 Y463D was found in both species, being detected in five and six isolates of *Mf* and *Mm*, respectively (Figure 2).

No CYP51 promoter modifications linked with overexpression, based on the insertion and duplication of a 19 bp repeat element with a palindromic core TAAATCTCGTACGATAGCA), were found in the nine *Mf* strains tested and the sensitive reference Bo-1 [17,33]. In contrast, the highly resistant *Mf* reference strain Ca10-6 harbored five extra copies of this element in the promoter, as previously described [10,25]. A single copy of an almost identical 19 bp element, TAAATCTCGTACGATTGCA (with a single base difference highlighted with underline), without replications was also present in the *CYP51* promoter (−118 bp upstream) of the 42 *Mm* strains.

### 3.2. Phenotypic Assessment of In Vitro Sensitivity to DMI Triazole Fungicides and Determination of EC_50_

Based on the average EC_50_ values, there were significant differences among the eight CYP51 variants detected in the population sampling of the BSDC pathogens (Table 3, Figure 2A,B, Appendix A). EC_50_ from variant G [T18I, V106D, and Y461D] from *Mf* was significantly higher for both DMI fungicides (EC_50_ propiconazole = 0.43 ± 0.30 µg mL^−1^; EC_50_ tebuconazole = 0.89 ± 0.03 µg mL^−1^), followed by variants F [T18I, V106D, and Y463D] (EC_50_ propiconazole = 0.29 ± 0.31 µg mL^−1^; EC_50_ tebuconazole = 0.73 ± 0.48 µg mL^−1^) and H [T18I, V106D, and G462D] (EC_50_ propiconazole = 0.27 ± 0.001 µg mL^−1^; EC_50_ tebuconazole = 0.86 ± 0.001 µg mL^−1^), also from *Mf*. Altogether, these three CYP51 variants from *Mf* had higher average EC_50_ values than all the variants detected in *Mm* isolates, which ranged from 0.01 to 0.07 µg mL^−1^ for propiconazole and from 0.03 to 0.37 µg mL^−1^ for tebuconazole (Table 3, Figure 2A,B).

For propiconazole, *Mm* strains harboring CYP51 variant A ([V106D and A446S]) showed sensitivity, whereas variants B ([V106D, Y136F, and A446S]), C ([V106D, A446S, and Y461H]), D ([V106D, A446S, Y461N, and Y463D]), and E ([V106D, A446S, and Y463D]) strains of *Mm* were classified as lowly resistant. *Mf* CYP51variants F, G, and H were all classified as moderately resistant. Regarding tebuconazole, CYP51 variants A strains B of *Mm* were classified as sensitive while variant B strains showed low levels of resistance. *Mm* CYP51 variant C, D, and E strains and *Mf* strains carrying CYP51 variants F, G, and H all showed moderate resistance levels to tebuconazole (Table 3). None of the BSDC isolates tested showed a highly resistant phenotype for either propiconazole and/or tebuconazole according to the resistance level categories presented in Table 2.

There were also significant differences among geographical populations of the pathogens (Table 3, Figure 2C,D, Appendix A). The population SPVR_CI with *Mf* isolates showed significantly higher levels of resistance to both fungicides than all the other three *Mm* populations sampled (MGN-C, SPNW-C, SPNW-O). The comparison between species indicated that *Mf* (EC_50_ propiconazole = 0.33 ± 0.29 µg mL^−1^; EC_50_ tebuconazole = 0.80 ± 0.36 µg mL^−1^) was significantly more resistant than *Mm* (EC_50_ propiconazole = 0.02 ± 0.05 µg mL^−1^; EC_50_ tebuconazole = 0.10 ± 0.18 µg mL^−1^) for both DMI fungicides (Table 3, Figure 2E,F). Among the *Mm* populations, population SPNW-C contained the least DMI-sensitive isolate belonging to CYP51 variant E together with sensitive isolates carrying variant A. Population MGN-C contained the remaining CYP51 variant E isolates and isolates carrying variants C and D, all being lowly resistant or moderately resistant to propiconazole and tebuconazole, respectively. Only isolates that were sensitive to propiconazole and sensitive and/or lowly resistant to tebuconazole were detected in population SPNW-O (Figure 2).

The linear model in Figure 3 describes the correlation between the DMI fungicides propiconazole and tebuconazole and the EC_50_ values from 51 isolates of *Mycosphaerella* spp. The coefficient of determination (*R*^2^ = 0.458) indicates a moderate correlation, suggesting that the linear model explains a portion of the variability in the data. The root mean square error (*RMSE* = 0.330) reflects the predictive capacity of the model. Based on this linear relationship, we detected significant (*p* < 0.001) cross-resistance between propiconazole and tebuconazole for *Mycosphaerella* spp. isolates.

## 4. Discussion

Resistance to the DMI fungicides propiconazole and tebuconazole was prevalent among the populations of *Mf* and *Mm* examined in banana plantations from Southeastern Brazil, with all 9 *Mf* strains and 11 out of 42 *Mm* strains tested showing different levels of resistance. These strains carried a range of CYP51 alterations, F136Y, Y461D/H/N, G462D, and Y463D, all known to affect azole binding (Figure 2, Table 3). The evolution of DMI target site resistance in *Mf* and *Mm* indicates that the frequently sprayed DMI fungicides, within the BSDC management system, exerted a high selection pressure on pathogen populations [24,28]. This underscores the urgent need to implement integrated disease management strategies aimed at reducing exclusive reliance on DMI fungicides and incorporating less aggressive practices to mitigate and slow down resistance development [19,20,36].

It is important to highlight that all isolates in this study were also analyzed for resistance to QoI and SDHI fungicides in previous studies, and many exhibited multiple resistance [19,20,37]. For example, out of the nine *Mf* isolates examined, one showed also resistance to the QoI fungicides azoxystrobin and trifloxystrobin [19,37], while all demonstrated reduced sensitivity to the SDHI fungicides boscalid and fluxapyroxad [20]. Regarding *Mm,* among the 42 isolates evaluated, seven exhibited resistance to QoI fungicides, and all isolates showed some degree of resistance to SDHI fungicides [20].

The disease management system for the BSDC may have contributed to the significant differences among the populations sampled regarding their resistance levels to DMI fungicides. For instance, the *Mf* population from Vale do Ribeira (SPVR_CI) exhibited significantly higher levels of resistance to DMI fungicides compared with the three sampled *Mm* populations. As similar *CYP51* mutations have evolved in *Mm* and *Mf* populations, additional resistance mechanisms might have evolved in *Mf* populations. It is noteworthy that banana plantations in Vale do Ribeira are subject to more intense fungicide spraying, with 8 to 14 annual preventive applications of fungicides for black Sigatoka control [19,20,37]. In contrast, banana plantations from the other sampled regions, such as Northwest São Paulo (SPNW-C) and Northern Minas Gerais (MGN_C), are subjected to less intensive fungicide management systems (where fungicide use is reduced to four or five preventive sprays for Sigatoka control), including even no fungicide use (SPNW_O), probably resulting in the absence of isolates with moderate resistance levels in the latter population (Figure 2). These findings underscore the significant influence of the fungicide management system on the evolution and spread of fungicide resistance within the BSDC [20] while also suggesting the presence of other contributing factors to this complex dynamic. These factors may include specific agricultural practices, genetic characteristics of local fungal populations, and multifaceted interactions among fungi, host plants, and the surrounding environment [36].

A total of 28 different amino acid alterations present in 60 different variants of the CYP51 protein have been identified in *Mf* strains sampled globally [25]. CYP51 alterations Y136F, A313G, H380N, A381G, D460E/V, ΔY461, Y461D/N/S, G462A/D, and Y463D/H/N/S have all been linked to DMI resistance. Other alterations, including T18I, A19E, Y59F, V106D, V117L, R416G, and A446S have been found in DMI-sensitive isolates and seem to have evolved at random as they are not located near substrate recognition sites or the haem-binding domain of the CYP51 protein [15,28]. In this study, we identified CYP51 substitutions T18I, V106D, Y461D, G462D, and Y463D in the nine *Mf* isolates originating from the Ribeira Valley in Southern São Paulo (SPVR-CI) (Figure 1, Table 3). CYP51 V106D seems to be present in all *Mf* and *Mm* strains that were globally sampled and tested so far [10,14,25]. The equivalent residue at position 107, D107, is also conserved in CYP51 of the closely related fungus *Zymoseptoria tritici* (aka *Mycosphaerella graminicola*), which also shares 81% amino acid sequence similarity. Therefore, it is likely that a rare random mutation, resulting in D106V, has evolved in the reference strain CIRAD86, as was also suggested in an earlier study [15]. CYP51 substitution T18I has been found at high frequencies in *Mf* populations sampled in Columbia, Ecuador, and the Philippines but seems to be absent in Cameroon. For *Mf*, CYP51 substitution A446S was exclusively found in the majority of isolates sampled in the Philippines, most of them also carrying mutations associated with resistance. A 3D model of CYP51 from *Mf* has shown that amino acid residues at positions 18, 106, and 446 are not near to any of the six substrate recognition sites (SRSs) or the haem-binding cavity and, therefore, are not involved in azole binding. However, an additive effect on DMI insensitivity of the A446S substitution in combination with other mutations cannot be ruled out [25]. CYP51 substitutions G462D and Y463D have been reported previously in *Mf* strains from Brazil, and these alterations together with Y461D are also found globally in *Mf* populations, often in combination with other mutations [10,14,25]. These mutations are located on a region of the protein at the haem end that is specific for fungi. Molecular modeling and azole docking studies on CYP51 from *Z. tritici* has shown that substitutions at positions 461–463 cause azole resistance by moving key residues V134 and or Y136, located on the access channel end, further away from the binding pocket. However, loss of the tyrosine residue at position 463 has probably the greatest impact on azole binding [38].

Mutations resulting in CYP51 substitutions V106D, Y136F, A446S, Y461D/H/N, and Y463D were detected in the 42 *Mm* strains. CYP51 V106D and A446S have been found before in Brazilian *Mm* populations and are not associated with DMI resistance [24]. Y136F, Y461D/N, and Y463D have also evolved in *Mf* populations, showing a parallel evolution of CYP51 in these closely related pathogens under selection of DMI fungicides. CYP51 alterations Y461N and the combination of A381G and Y463H have been reported previously for Brazilian strains [6,14,20]. CYP51 Y461H in *Mm* is a novel discovery, reported for the first time here and, so far, not found in any BSDC population that has been sampled and tested so far. CYP51 Y136F was detected for the time in *Mm* as part of this study. A substitution at Y136 (Y137 in *Z. tritici*), or its equivalent in other species, is the most frequently observed modification of CYP51 in pathogenic fungi [28]. V136 and Y137 are located on the access channel end of the binding pocket. Molecular modeling and azole docking studies on *Z. tritici* CYP51 shows that the mutant F137 residue is pushed into a position preventing the proper binding of triadimenol, which explains the observed high levels of resistance in corresponding *Z. tritici* field isolates [38]. However, the same study predicted a much smaller influence of the Y137F substitution on the binding of other azole agents. This also explains the low levels of resistance to propiconazole and tebuconazole in this study and support the hypothesis that this mutation was likely selected in *Mm* populations after exposure to triadimenol in the past.

The analysis of mutation combinations present in the *Mf* and *Mm* isolates allowed the identification of eight distinct CYP51 protein variants. These CYP51 variants containing between 2 and 4 amino acid alterations were designated as follows: (A) [V106D and A446S] (*Mm*, N = 33); (B) [V106D, Y136F, and A446S] (*Mm*, N = 2); (C) [V106D, A446S, and Y461N] (*Mm*, N = 3); (D) [V106D, A446S, Y461H, and Y463D] (*Mm*, N = 1); (E) [V106D, A446S, and Y463D] (*Mm*, N = 3); F) [T18I, V106D, and Y463D] (*Mf*, N = 5); (G) [T18I, V106D, and Y461D] (*Mf*, N = 3) and (H) [T18I, V106D, and G462D] (*Mf*, N = 1). These haplotypes highlight significant mutation diversity, revealing a complex genetic landscape within these fungal populations, but even more complex CYP51 haplotypes have evolved in *Mf* populations under strong selection pressure by DMI fungicides, with up to six amino acid substitutions reported for some variants (e.g., [V106D, K171R, A313G, A446S, D460E, and Y461N] and [T18I, V106D, Y136F, A313G, A381G, and Y461D]) present in populations sampled in the Philippines and Costa Rica [10,25].

Recently, another mechanism of DMI resistance in *Mf* was identified, involving overexpression of the *CYP51* gene due to the in tandem insertion of small repetitive elements in its promoter region [17]. These repetitive sequences were also transmitted across generations of the fungal pathogen [10]. Strains of *Mf* sampled in Africa, Asia, and South America showing complex CYP51 variants in combination with the presence of up to six repetitive elements in their promoter, thereby enhancing the expression up to five-fold, were found to show high levels of resistance to different DMI fungicides [17,28,33]. However, *CYP51* promoter inserts were not detected in the BSDC isolates examined in this study, and differences in sensitivity to propiconazole and tebuconazole between the two species cannot be explained by this mechanism. MDR (multidrug resistance) due to overexpression of efflux pumps was already detected in a selection of *Mf* and *Mm* isolates examined in this study previously [20].

In summary, the current study represents a significant contribution to an improved understanding of the mechanisms, emergence, and spread of DMI fungicide resistance in BSDC populations in Brazil. It also underscores the impact of extensive use of chemical fungicides and the associated risks of further selecting and dispersing BSDC strains with higher levels of DMI resistance [19,20,36]. To mitigate these risks, it is imperative to consider innovative disease management approaches that minimize reliance on preventive fungicide sprays with medium to high resistance risk [36].

One proposed solution to fungicide resistance involves implementing a smart spraying system that is not based on fixed spray schedules but rather on real-time information about the temporal dynamics of airborne inoculum of these phytopathogens (i.e., aerobiology data) and epidemic risks [36,39,40]. This system would enable more precise and timely decision-making on when and what to spray. For example, if the presence of fungal inoculum with resistance alleles to major fungicide groups such as DMIs is detected, the system would alternatively recommend spraying only low-resistance risk active ingredients, thereby reducing further selective pressure on populations and the likelihood of resistance fixation [36].

## 5. Conclusions

DMI fungicide resistance prevalence: Widespread resistance to DMI fungicides was observed among BSDC populations in banana plantations of Southeastern Brazil. All nine *Mf* isolates showing moderate resistance levels to propiconazole or tebuconazole, while 26% of the *Mm* strains (11 out of 42) tested showed low to moderate levels of resistance to these DMI fungicides.

Mutation diversity and resistance association: Mutations leading to CYP51 substitutions Y136F, lY461N/H, and Y463D in *Mm* and Y461D, G462D, and Y463D in *Mf* were associated with low or moderate levels of resistance to DMIs. More complex CYP51 variants and *CYP51* promoter inserts associated with upregulation of the target protein were not detected and can explain the absence of highly DMI-resistant strains in Brazil.

Influence of fungicide management system: Variations in resistance levels were strongly influenced by the intensity of fungicide management practices. The *Mf* population from Vale do Ribeira exhibited notably higher resistance due to more intense fungicide spraying compared with regions with less intensive fungicide use.

Recommendations for integrated disease management: Implement integrated disease management strategies to reduce reliance on fungicides and adopt practices to mitigate resistance development. Develop and deploy smart spraying systems guided by real-time aerobiology data to optimize fungicide application and preserve efficacy.

## Figures and Tables

**Figure 1 microorganisms-13-01439-f001:**
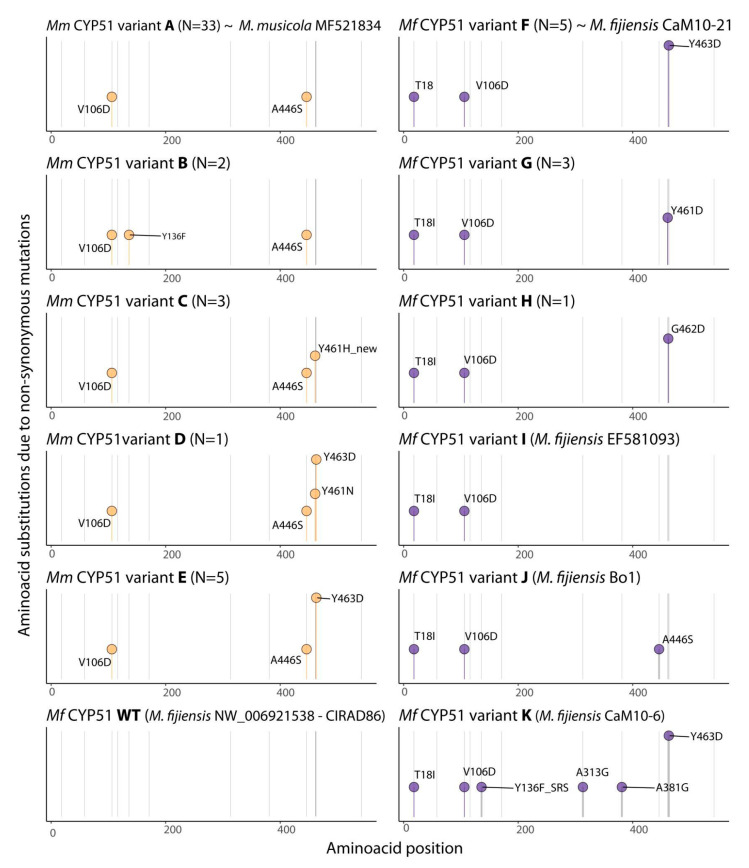
Lollipop depiction of non-synonymous amino acid substitutions conferred by mutations in the *CYP51* gene of isolates of *Mycosphaerella musicola* (at **left**) and *Mycosphaerella fijiensis* (at **right**) sampled from four geographical populations from distinct fungicide management systems in Minas Gerais and São Paulo, Brazil. CYP51 variants A to E were only detected in isolates of *Mm*, while CYP51 variants F, G, and H were found in *Mf* isolates. Orange lollipops indicate amino acid substitutions in the inferred CYP51 sequence for *Mm* while purple lollipops mark CYP51 substitutions found in *Mf*. When the protein variant detected has shown similarity with reference sequences from the GenBank/NCBI, the sequence code is indicated at the respective variant (e.g., CYP51 variant A was identical to *Mm* MF521834; variant F was identical to the sequence from *Mf* CaM10-21). CYP51 WT and variants I, J, and K were not detected in our study.

**Figure 2 microorganisms-13-01439-f002:**
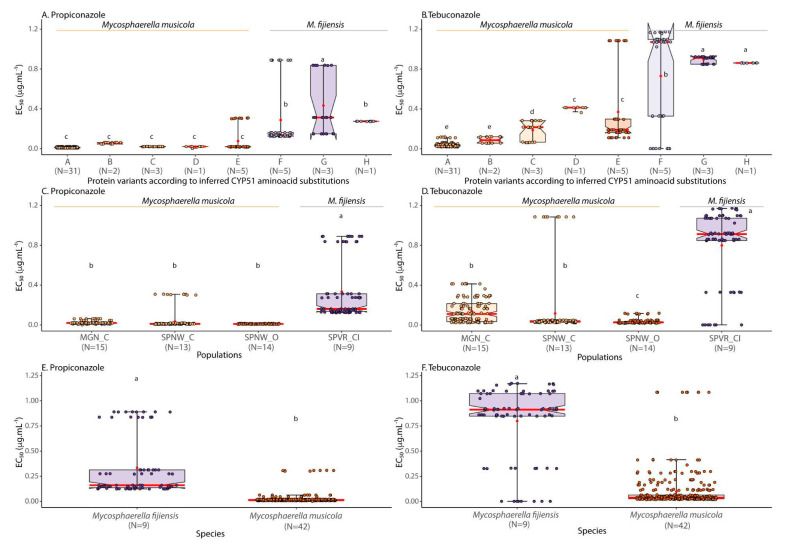
Boxplot comparing the effects of CYP51 protein variants, geography, and species in the levels of sensitivity (EC_50_ in µg mL^−1^) to DMI fungicides propiconazole and tebuconazole in BSDC populations sampled from banana plantations in São Paulo and Minas Gerais, Brazil. Red dots indicate the average EC_50_ values for each category. Means followed by the same lowercase letter are not significantly different by the ScottKnott test at 5% probability.

**Figure 3 microorganisms-13-01439-f003:**
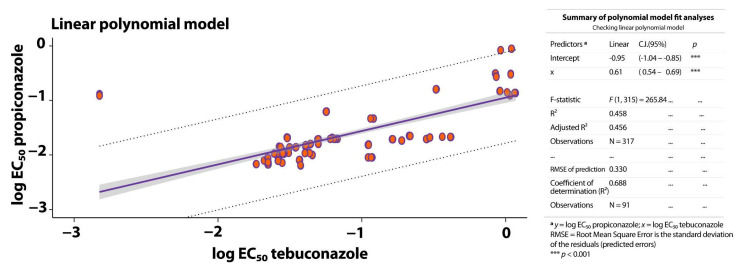
Correlation between the log EC_50_ data for sensitivity of the BSDC pathogens *Mycosphaerella* spp. (N = 51) to the DMI fungicides propiconazole and tebuconazole. The scatterplot illustrates the relationship between propiconazole concentrations (*y*-axis) and tebuconazole concentrations (*x*-axis) and the corresponding EC_50_ values. The purple lines represent the linear regression model.

**Table 1 microorganisms-13-01439-t001:** Specific primers used for PCR amplification aimed at species identification and for Sanger sequencing to determine allelic variation in the *CYP51* gene in *Mycospharella fijiensis* and *M. musicola* *.

Primers (Source) **	Sequence (5′-3′)	Species	Annealing Temperature (°C)
CYP51_Pfijien_F1 [10,33]	AAGGTCATATCGCAGG	*Mf*	56
CYP5_1Mf592F (S)	TGAATGGAAAGCTCAAGGACGT	*Mf*	58
CYP51_Mf1765R (S)	ACTTCTCACTTGGGTTCTCGTC	*Mf*	58
CYP51_Pfijien_R1 [10,33]	GAATGTTATCGTGTGACA	*Mf*	56
CYP51A_Mm_F523	ATCGACATCACCACCGCTG	*Mm*	59
CYP51A_Mm_R1554 (S)	TGTCCTCCTCTTTCTCTCTCCAGA	*Mm*	59
CYP51A_Mm_F1369 (S)	TTGCCGATCTCTACCACGAC	*Mm*	59
CYP51A_Mm_R2427-2	CGCATGACAGATAAAGAGGAAAGC	*Mm*	59
CYP51_Prom_1800_F	GAA CGA GTT TCC AGG TTG CTG	*Mm*	58
CYP51_Prom_2082_R	GCA GTT TGT GTG AAA GCA GGG	*Mm*	58

* Because the *CYP51* gene is relatively long in both pathogens, multiple pairs of primers were used (two pairs for *Mf* and three pairs for *Mm*) to fully cover the gene and the promoter region. ** Unless otherwise indicated, the primers were specifically designed for this study.

**Table 2 microorganisms-13-01439-t002:** Sensitivity classes for phenotypic classification of *Mycosphaerella fijiensis* and *M. musicola* isolates to the DMI fungicides propiconazole and tebuconazole based on EC_50_ values.

Phenotypic Classes	EC_50_ Intervals (µg mL^−1^)	
	**Propiconazole**	**Tebuconazole**

Sensitive (S)	≤0.01	≤0.05
Lowly Resistant (LR)	>0.01 to 0.1	>0.05 to 0.1
Moderately Resistant (MR)	>0.1 to 1.0	>0.1 to 1.0
Highly Resistant (HR)	>1.0	>1.0

**Table 3 microorganisms-13-01439-t003:** Summary outlining five specific substitutions and their corresponding CYP51 protein variants and impact on resistance levels to DMI fungicides in *Mycosphaerella fijiensis* (*Mf*) and *M. musicola* (*Mm*) from the BSDC in Southeastern Brazil *.

Species	CYP51 Variant ^1^	Alterations Associated with Resistance	Average EC_50_ Propiconazole (µg mL^−1^)	Propiconazole Resistance Category ^2^	Average EC_50_ Tebuconazole (µg mL^−1^)	Tebuconazole Resistance Category ^2^
*Mm*	A (31)	-	0.01 c	S	0.03 e	S
*Mm*	B (2)	Y136F	0.05 c	LR	0.09 e	LR
*Mm*	C (3)	Y461H	0.02 c	LR	0.19 d	MR
*Mm*	D (1)	Y461N, Y463D	0.02 c	LR	0.41 c	MR
*Mm*	E (5)	Y463D	0.08 c	LR	0.37 c	MR
*Mf*	F (5)	Y463D	0.29 b	MR	0.73 b	MR
*Mf*	G (3)	Y461D	0.43 a	MR	0.89 a	MR
*Mf*	H (1)	G462D	0.27 b	MR	0.86 a	MR

^1^ CYP51 variant with number of strains between brackets; ^2^ resistance categories are S (sensitive), LR (lowly resistant), and MR (moderately resistant). * Average EC50 values followed by the same lowercase letter are not significantly different by the ScottKnott test at 5% probability.

## Data Availability

The *CYP51* experimental sequence data from *Mf* and *Mm* populations sampled in Southeastern Brazil and that supports the findings of allelic variation in the target genes for demethylation inhibitor (DMI) triazole fungicides sensitivity will be available at the GenBank/NCBI database. Upon publication, the complete data set of phenotypic data presented in this study will be publicly available at the Mendeley Data repository.

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
