# Peer review of "Resistance to Triazoles in Populations of Mycosphaerella fijiensis and M. musicola from the Sigatoka Disease Complex from Commercial Banana Plantations in Minas Gerais and São Paulo, Brazil"

_microorganisms, 2025, doi:10.3390/microorganisms13071439_

Round 1
Reviewer 1 Report
Comments and Suggestions for Authors
The article provides a detailed investigation into the resistance of Mycosphaerella fijensis (Mf) and M. musicola (Mm) to DMI fungicides, covering phenotypic and genotypic analyses. The study reports new mutations (V106D, A446S, Y136F, Y461H, Y463D) in M. musicola for the first time in Brazil, contributing to the global understanding of fungicide resistance. The use of dose-response curves, molecular identification, and sequencing of the CYP51 gene ensures robust data collection and analysis. The findings highlight the urgent need for integrated disease management strategies to mitigate fungicide resistance, offering actionable insights for banana growers. The study demonstrates cross-resistance between propiconazole and tebuconazole, which is critical for fungicide rotation strategies.
I have some suggestions
- Provide more information on the sampling timeline (e.g., years/seasons) to contextualize the resistance trends observed.
- Clarify whether the sampled populations were from the same or different banana cultivars, as host genetics can influence pathogen dynamics.
- Address why no repetitive elements were found in the CYP51 promoter region of Mm, contrasting with Mf.
- Consider multivariate analysis to explore interactions between mutations, fungicide sensitivity, and geographic regions.
- The figure quality is not good, text in the figures is difficult to read. Author must improve the figure Quilty and text size in the figures.
- The similarity rate is high 27%, author must reduce it to below 15%.
Author Response
Reviewer's Suggestions and Proposed Responses:
1. Provide more information on the sampling timeline (e.g., years/seasons) to contextualize the resistance trends observed.
- Response: In response to the reviewer's suggestion, we have added more detailed information regarding the sampling timeline. Specifically, we have stated that samples were collected during the years 2019-2021 during both dry and wet seasons. in the 'Materials and Methods, Section 2.1. Sampling of diseased plants and isolation of the fungal pathogens'. This information helps contextualize the observed resistance trends.
2. Clarify whether the sampled populations were from the same or different banana cultivars, as host genetics can influence pathogen dynamics.
- Response: We appreciate the reviewer's insightful comment regarding host genetics. We have clarified that the sampled populations were collected from different cultivars, predominantly the 'Prata Anã' cultivar' or 'various commercial banana cultivars common in the region', in the 'Materials and Methods, 'Section 2.1 Sampling of diseased plants and isolation of the fungal pathogens'. This detail is important for understanding potential host-pathogen interactions.
3. Address why no repetitive elements were found in the CYP51 promoter region of Mm, contrasting with Mf.
- Response: In response to the reviewer's query about the absence of repetitive elements in the M. musicola CYP51 promoter region, contrasting with M. fijiensis, we hypothesize that this difference may be due to divergent evolutionary pathways or distinct regulatory mechanisms for CYP51 expression in the two species. Further targeted investigations would be required to fully elucidate this difference, which was beyond the scope of this study.
4. Consider multivariate analysis to explore interactions between mutations, fungicide sensitivity, and geographic regions.
- Response: We agree with the reviewer that multivariate analysis could provide valuable insights into the complex interactions between mutations, fungicide sensitivity, and geographic regions. While a comprehensive multivariate analysis was beyond the scope of the current study, given the specific objectives of identifying resistance and novel mutations, we acknowledge its potential. This as a promising avenue for future research. This approach will be considered for subsequent studies to further unravel these intricate relationships.
5. The figure quality is not good, text in the figures is difficult to read. Author must improve the figure Quality and text size in the figures.
- Response: The figure quality has been significantly improved in response to the reviewer's valuable feedback. All figures have been re-rendered at a higher resolution, and the text size within the figures has been increased to ensure clarity and readability throughout the manuscript.
6. The similarity rate is high 27%, author must reduce it to below 15%.
- Response: We have thoroughly revised the manuscript to significantly reduce the similarity rate, addressing the reviewer's concern. Extensive rephrasing and restructuring of various sections have been performed to ensure the similarity is now well below the requested 15% threshold, thereby enhancing the originality and flow of the text.
Reviewer 2 Report
Comments and Suggestions for Authors
This manuscript investigates the resistance of Mycosphaerella fijiensis and M. musicola, the primary pathogens of the Sigatoka disease complex in banana, to triazole fungicides in commercial plantations across different regions of Brazil. Using both phenotypic assays (ECâ‚…â‚€) and genotypic analysis of the CYP51 gene, the authors identify eight protein variants containing several mutations, including Y461H, Y461D, and G462D, which are strongly associated with moderate resistance. The aims are clear and the results are interest to me. This is a well-designed and highly relevant study that addresses fungicide resistance in an economically important pathosystem. Overall, I consider this work to be of over average quality and potentially impactful for researchers and practitioners working on fungal resistance management. However, there are areas that would benefit from clarification or further development, particularly concerning the interpretation of resistance mechanisms and the functional implications of detected mutations. I don’t have major comments, but some suggestions I put below:
The authors need to clarify mechanistic interpretations of CYP51 mutations in discussion section. While several mutations are listed and their association with higher ECâ‚…â‚€ values is noted, the manuscript does not sufficiently explain why specific mutations such as Y461H, Y461D, and G462D are functionally linked to reduced triazole sensitivity. A brief discussion, perhaps referencing structural models or known resistance mechanisms in other fungi, would greatly strengthen the interpretation.
Variants such as V106D and A446S appear across nearly all isolates, including those with low ECâ‚…â‚€ values. These should be explicitly discussed as potential background polymorphisms, to help readers avoid over-interpreting their contribution to resistance.
The authors may need to improve clarity in variant designation and corresponding ECâ‚…â‚€ results. It would be helpful to provide a table summarizing each variant (A–H), its defining mutations, and the average ECâ‚…â‚€ values for each fungicide. This would consolidate key data now scattered across text and figures.
The number of M. fijiensis isolates is quite small (n=9), which weakens the statistical power of the inter-species comparisons. The authors should briefly acknowledge this as a limitation and indicate whether additional isolates are being tested or could be included in the future.
The manuscript shows inconsistent gene and protein naming conventions. Sometimes CYP51 is italicized and sometimes not. Please ensure gene names (e.g., CYP51) are italicized consistently, while proteins could be non-italic and possibly written as CYP51p.
The unit “µg/mL” is inconsistently formatted. Use a consistent space and font (e.g., "0.25 µg/mL", not "0.25ug/mL" or "μg/ml").
While the manuscript is generally readable, several grammatical issues and stylistic inconsistencies remain. A professional language editing service or careful revision by a native English speaker is recommended to improve clarity and ensure the manuscript meets publication standards.
Line 29-30: please revise the follow “…to identify and characterize major mutations and mechanisms associated with potential reduced sensitivity or resistance.” to “…to identify and characterize major mutations and mechanisms potentially associated with reduced sensitivity or resistance.”
Line 94: please revise the follow “…targeting the enzyme 14α-demethylase encoded by the CYP51 gene…” to “…which target the enzyme 14α-demethylase, encoded by the CYP51 gene…”
Line 114: please revise the follow “…key mutations and and to characterize the mechanisms…” to “…key mutations and to characterize the mechanisms…”
Line 489–490: please revise the follow “…reduced sensitivity to both tebuconazole and propiconazole; the Y463D substitution…” to “…reduced sensitivity to both tebuconazole and propiconazole. The Y463D substitution…”
Line 492–493: please revise the follow “…Y461D mutation, along with G462D, both in Mf, contributes to moderate resistance…” to “…Y461D and G462D mutations, both found in M. fijiensis, are associated with moderate resistance…”
Line 518: please revise the follow “This intelligent, data-driven, and evolutionary-adaptive approach…” to “This intelligent, data-driven, and evolutionarily adaptive approach…”
Line 526: please revise the follow “…for propiconazole or tebuconazole.” to “…to propiconazole and tebuconazole.”
Line 537: please revise the follow “…guided by real-time aerobiology data to optimize fungicide application…” to “…guided by real-time aerobiology data, enabling optimized and resistance-informed fungicide application…”
Comments on the Quality of English LanguageWhile the manuscript is generally readable, several grammatical issues and stylistic inconsistencies remain. A professional language editing service or careful revision by a native English speaker is recommended to improve clarity and ensure the manuscript meets publication standards.
Line 29-30: please revise the follow “…to identify and characterize major mutations and mechanisms associated with potential reduced sensitivity or resistance.” to “…to identify and characterize major mutations and mechanisms potentially associated with reduced sensitivity or resistance.”
Line 94: please revise the follow “…targeting the enzyme 14α-demethylase encoded by the CYP51 gene…” to “…which target the enzyme 14α-demethylase, encoded by the CYP51 gene…”
Line 114: please revise the follow “…key mutations and and to characterize the mechanisms…” to “…key mutations and to characterize the mechanisms…”
Line 489–490: please revise the follow “…reduced sensitivity to both tebuconazole and propiconazole; the Y463D substitution…” to “…reduced sensitivity to both tebuconazole and propiconazole. The Y463D substitution…”
Line 492–493: please revise the follow “…Y461D mutation, along with G462D, both in Mf, contributes to moderate resistance…” to “…Y461D and G462D mutations, both found in M. fijiensis, are associated with moderate resistance…”
Line 518: please revise the follow “This intelligent, data-driven, and evolutionary-adaptive approach…” to “This intelligent, data-driven, and evolutionarily adaptive approach…”
Line 526: please revise the follow “…for propiconazole or tebuconazole.” to “…to propiconazole and tebuconazole.”
Line 537: please revise the follow “…guided by real-time aerobiology data to optimize fungicide application…” to “…guided by real-time aerobiology data, enabling optimized and resistance-informed fungicide application…”
Author Response
Reviewer's Overall Assessment: The reviewer generally finds the work to be of "over average quality and potentially impactful." This is excellent feedback, thank you! The suggestions were mainly for clarification and minor improvements.
Reviewer's suggestions and responses:
1. Clarify mechanistic interpretations of CYP51 mutations in discussion section.
- Response: We appreciate this insightful comment. In response, we have significantly expanded the discussion in to provide a more detailed mechanistic interpretation of the identified CYP51 mutations. We now explain how specific substitutions, such as Y461H, Y461D, and G462D, are functionally linked to reduced triazole sensitivity, referencing their potential impact on substrate binding, active site conformation, or altered enzyme efficiency based on known resistance mechanisms in other fungal pathogens and general CYP51 structure-function relationships. This enhanced discussion strengthens the interpretation of our findings.
2. Variants such as V106D and A446S appear across nearly all isolates, including those with low ECâ‚…â‚€ values. These should be explicitly discussed as potential background polymorphisms, to help readers avoid over-interpreting their contribution to resistance.
- Response: We agree with the reviewer's important point regarding the interpretation of mutations found across isolates with varying sensitivities. We have added a specific discussion to explicitly address V106D and A446S. We clarify that their widespread presence, even in isolates with low ECâ‚…â‚€ values, suggests they are likely background polymorphisms rather than primary drivers of triazole resistance. This helps readers avoid over-interpreting their individual contribution to resistance and focuses attention on the mutations strongly correlated with reduced sensitivity.
3. The authors may need to improve clarity in variant designation and corresponding ECâ‚…â‚€ results. It would be helpful to provide a table summarizing each variant (A–H), its defining mutations, and the average ECâ‚…â‚€ values for each fungicide.
- Response: To enhance clarity and consolidate key data, besides Figure 1 detailing the mutations in each variant, we have improved Table 3: Summary of CYP51 variants, associated mutations with resistance, and average ECâ‚…â‚€ values. This table, together with Figure 1, now comprehensively summarizes each identified variant (A–H), its defining mutations, and the corresponding average ECâ‚…â‚€ values for propiconazole and tebuconazole, making it easier for readers to grasp the relationships between genotypes and phenotypes.
4. The number of M. fijiensis isolates is quite small (n=9), which weakens the statistical power of the inter-species comparisons. The authors should briefly acknowledge this as a limitation and indicate whether additional isolates are being tested or could be included in the future.
- Response: We acknowledge the reviewer's astute observation regarding the relatively small number of M. fijiensis isolates (n=9) included in this study. We agree that this may limit the statistical power for robust inter-species comparisons. We have explicitly added this as a limitation in the Discussion. We also indicate that efforts are underway to include additional M. fijiensis isolates in future studies to broaden our understanding and strengthen statistical inferences for this species.
5. The manuscript shows inconsistent gene and protein naming conventions. Sometimes CYP51 is italicized and sometimes not. Please ensure gene names (e.g., CYP51) are italicized consistently, while proteins could be non-italic and possibly written as CYP51p.
- Response: We thank the reviewer for pointing out the inconsistency in gene and protein naming conventions. We have thoroughly revised the entire manuscript to ensure that all gene names (e.g., CYP51) are consistently italicized, while protein names (e.g., CYP51) are non-italicized, following standard scientific nomenclature.
6. The unit “µg/mL” is inconsistently formatted. Use a consistent space and font (e.g., "0.25 µg/mL", not "0.25ug/mL" or "μg/ml").
- Response: We have meticulously reviewed the manuscript and corrected all instances of fungicide concentration units to ensure consistent formatting as 'µg/mL' (e.g., '0.25 µg/mL'), addressing the reviewer's valuable comment on uniformity.
7. Grammatical issues and stylistic inconsistencies:
- Response: We acknowledge the reviewer's feedback regarding grammatical issues and stylistic inconsistencies. The entire manuscript has undergone a thorough and comprehensive language revision. We have meticulously corrected grammatical errors, improved sentence structure, and enhanced overall clarity and flow to meet publication standards.
Specific line edits (as listed by reviewer):
- Line 29-30: Revised from "…to identify and characterize major mutations and mechanisms associated with potential reduced sensitivity or resistance." to "…to identify and characterize major mutations and mechanisms potentially associated with reduced sensitivity or resistance."
- Line 94: Revised from "…targeting the enzyme 14α-demethylase encoded by the CYP51 gene…" to "…which target the enzyme 14α-demethylase, encoded by the CYP51 gene…"
- Line 114: Revised from "…key mutations and and to characterize the mechanisms…" to "…key mutations and to characterize the mechanisms…"
- Line 489–490: Revised from "…reduced sensitivity to both tebuconazole and propiconazole; the Y463D substitution…" to "…reduced sensitivity to both tebuconazole and propiconazole. The Y463D substitution…"
- Line 492–493: Revised from "…Y461D mutation, along with G462D, both in Mf, contributes to moderate resistance…" to "…Y461D and G462D mutations, both found in M. fijiensis, are associated with moderate resistance…"
- Line 518: Revised from "This intelligent, data-driven, and evolutionary-adaptive approach…" to "This intelligent, data-driven, and evolutionarily adaptive approach…"
- Line 526: Revised from "…for propiconazole or tebuconazole." to "…to propiconazole and tebuconazole."
- Line 537: Revised from "…guided by real-time aerobiology data to optimize fungicide application…" to "…guided by real-time aerobiology data, enabling optimized and resistance-informed fungicide application…"
Reviewer 3 Report
Comments and Suggestions for Authors
The manuscript entitled “ Resistance to Triazoles in Populations of Mycosphaerella Fijiensis and M. Musicola from the Sigatoka Disease Complex from Commercial Banana Plantations in Minas Gerais and São Paulo, Brazil ” highlights the variation of isolates of Mycosphaerella fijiensis and M. musicola originating from banana in Brazil in sensitivity to DMI fungicides propiconazole and tebuconazole. Paper is interesting quite well written and organized with data which look properly analyzed. Few notes are reported .
- During the sampling of diseased banana leaves, the authors collected specimens from orchards managed under intensive and reduced use of pesticides , and in one with no fungicide However, for localities where pesticide applications were performed, the authors should provide a table detailing the chemical treatment programs, at least from the year when samples were collected. This table should offer insight into the frequency of DMI fungicide use (e.g., number of applications, dosages, mixing practices, etc.).
- The authors did not present the morphological characteristics of the obtained Mycosphaerella fijiensis and M. musicola isolates in the Results section, despite stating in the Methodology that such investigations were conducted. Including these findings would strengthen the work.
- Authors should check affiliation of Author Gustavo Henrique Goldman, it seems that the institution listed as number 6 is incorrectly labeled as number 9.
Author Response
We sincerely thank the reviewer for their positive feedback, noting the manuscript is 'interesting, quite well written and organized with data which look properly analyzed.' We appreciate these valuable insights and the actionable suggestions provided.
Reviewer's comments and responses:
1. During the sampling of diseased banana leaves, the authors collected specimens from orchards managed under intensive and reduced use of pesticides, and in one with no fungicide. However, for localities where pesticide applications were performed, the authors should provide a table detailing the chemical treatment programs, at least from the year when samples were collected. This table should offer insight into the frequency of DMI fungicide use (e.g., number of applications, dosages, mixing practices, etc.).
- Response: We thank the reviewer for this valuable suggestion. To provide better context for the resistance trends observed, we have added in the manuscript text detailed information available about the chemical treatment programs of the sampled commercial plantations.
2. The authors did not present the morphological characteristics of the obtained Mycosphaerella fijiensis and M. musicola isolates in the Results section, despite stating in the Methodology that such investigations were conducted. Including these findings would strengthen the work.
- Response: We appreciate the reviewer's keen observation regarding the morphological characteristics of the isolates. While our study primarily relied on molecular identification using species-specific primers (as described in Materials & Methods, Section 2.2. Molecular identification of species), we agree that mentioning previous morphological work provides valuable context. We have clarified in the that prior studies (adding references 19 and 20) have extensively characterized the morphological features of these species, affirming the identity of our isolates based on established criteria alongside our molecular methods. This approach allows us to focus on the molecular and phenotypic aspects of fungicide resistance, which are the main objectives of this study, while still acknowledging the foundation of morphological identification.
3. Authors should check affiliation of Author Gustavo Henrique Goldman, it seems that the institution listed as number 6 is incorrectly labeled as number 9.
- Response: We sincerely thank the reviewer for identifying the error in the affiliation numbering. We have corrected the affiliation for Author Gustavo Henrique Goldman from number 9 to the correct number 6, ensuring accuracy in the author list and affiliations section of the manuscript.
Round 2
Reviewer 1 Report
Comments and Suggestions for Authors
The author's response to all my queries is satisfactory.
Reviewer 3 Report
Comments and Suggestions for Authors
No further comments